# Personalized Brain–Computer Interface and Its Applications

**DOI:** 10.3390/jpm13010046

**Published:** 2022-12-26

**Authors:** Yixin Ma, Anmin Gong, Wenya Nan, Peng Ding, Fan Wang, Yunfa Fu

**Affiliations:** 1Faculty of Information Engineering and Automation, Kunming University of Science and Technology, Kunming 650500, China; 2Brain Cognition and Brain-Computer Intelligence Integration Group, Kunming University of Science and Technology, Kunming 650500, China; 3School of Information Engineering, Chinese People’s Armed Police Force Engineering University, Xian 710086, China; 4Department of Psychology, College of Education, Shanghai Normal University, Shanghai 200234, China

**Keywords:** personalized brain–computer interface (pBCI), specific BCI user, customized brain–computer interface (cBCI), general brain–computer interface (gBCI), brain–computer interface (BCI)

## Abstract

Brain–computer interfaces (BCIs) are a new technology that subverts traditional human–computer interaction, where the control signal source comes directly from the user’s brain. When a general BCI is used for practical applications, it is difficult for it to meet the needs of different individuals because of the differences among individual users in physiological and mental states, sensations, perceptions, imageries, cognitive thinking activities, and brain structures and functions. For this reason, it is necessary to customize personalized BCIs for specific users. So far, few studies have elaborated on the key scientific and technical issues involved in personalized BCIs. In this study, we will focus on personalized BCIs, give the definition of personalized BCIs, and detail their design, development, evaluation methods and applications. Finally, the challenges and future directions of personalized BCIs are discussed. It is expected that this study will provide some useful ideas for innovative studies and practical applications of personalized BCIs.

## 1. Introduction

Brain–computer interfaces (BCIs) are designed to bypass users’ nerves and muscles to realize direct interaction between their brain and external devices through their brain signals. BCIs have potential medical [1] and non-medical applications. So far, researchers have conducted many studies on BCIs [1,2], proposed a general BCI system [3], and expect it to have a wide range of applications.

BCIs were first proposed by the University of California, Los Angeles (UCLA) in the 1970s [4]. At that time, experiments were conducted on animals to establish a direct communication path between the external environment (or equipment) and the brain [5,6]. Among them, two groundbreaking studies [7,8,9] published in Nature in 2012 showed how the BCI system achieved neuro arm control and arm motion recovery after paralysis. In 2017, Ajiboye et al. confirmed the effectiveness of the intracortical BCI system in the rehabilitation of patients with limb disorders [10]. BCI is ultimately driven or controlled by brain signals induced by a specific user’s mental activities. Although the connection between external devices and the brain has been established, and the availability of BCI systems can be verified from both invasive and non-invasive aspects, a BCI is ultimately driven or controlled by specific users’ psychological activities and brain signals induced by them. [11]. It is difficult for a general BCI system to meet the needs of specific users, so it is necessary for researchers and developers to customize personalized BCI systems for individuals. In recent years, to respond to the need of personalized therapy, the research on personalized and family therapy programs has expanded rapidly [12]. Personalized BCIs have gradually become an important research direction.

Personalized BCIs have been designed and developed on the basis of general BCIs, and customized BCI systems for specific BCI users according to their characteristics. In 2009, Borman et al. proposed a feature selection method to effectively reduce the dimension of multi-channel feature space, and select the feature most suitable for a specific user from the feature space [13]. In 2011, Arvaneh et al. proposed a sparse common spatial pattern (SCSP) algorithm for brain signal channel selection to select channels for specific users [14]. In 2015, Weyand et al. found that there were significant differences between subjects in the stimulation caused by the same mental task and cognitive process, and proposed a user-centered scoring method for mental tasks [15]. In 2017, Mastakouri et al. applied transfer learning to personalized BCIs to customize the brain signal classification model for specific users [16]. In 2020, Qi W et al. proposed a multi-mode wearable system for continuous and real-time breathing pattern monitoring during daily activities [17].

The above research covers the personalized development of BCI technology, that is, customizing BCI systems to meet the needs of specific users. However, although some aspects of personalized BCIs have been explored by a few studies [11,14,18,19], compared with general BCIs, there are fewer studies on them and there is no standardized definition. Personalized BCIs also have many problems that need to be further investigated. First, the paradigm design of personalized BCI is a difficult point. How to evaluate user satisfactions with the selected paradigm is also a problem and especially how to customize a satisfactory brain signal acquisition scheme for a specific user. Second, to improve the decoding performance of personalized BCI, what problems need to be solved in the personalized processing of brain signals and neurofeedback? At present, few studies have elaborated on these two aspects. Finally, the key to proving whether a system is effective is whether it can meet the relevant evaluation criteria. At present, there is no comprehensive evaluation method for personalized BCI performance. Whether the personalized BCIs proposed by researchers can meet the needs of specific users remains to be discussed.

In order to solve the above problems, it is necessary to design perfect evaluation criteria for personalized BCIs, propose a standard definition of personalized BCIs, and try to meet the needs of specific users in the process of designing personalized BCI systems.

To this end, this study will first give a definition of personalized BCIs, and then discuss the research and development of personalized BCIs. It is worth mentioning that this study attempts to propose a personalized BCI evaluation standard based on the evaluation indicators of the general BCI. Finally, this study introduces and discusses the application, challenges, limitations, and prospects of personalized BCIs.

## 2. Personalized BCI

A personalized BCI is designed and developed for a specific user on the basis of the general BCI. For this reason, before describing personalized BCI, we will first introduce the general BCI for comparison.

### 2.1. General BCI

Figure 1 is the schematic diagram of the general BCI system, showing the main parts of a BCI system [20]. Most BCI systems rely on three typical BCI paradigms: Motor Imagery (MI), P300, and steady-state visual-evoked potentials (SSVEP). At the same time, in the process of brain signal acquisition, we can collect electroencephalogram (EEG), near infrared spectroscopy (NIRS), and other signals through non-invasive devices, or use invasive devices to collect electrocartography (ECoG) [21,22] and local field potential(LFP) [23] signals. The personalized processing of brain signals includes preprocessing, feature extraction and selection, and classification. The classical algorithms for brain signal preprocessing include Kalman filtering [24] and the Independent Component Correlation Algorithm [25]. There are also many feature extraction and selection algorithms, such as covariance matrix [26] and the tensor method [27]. At present, there is more research on the classification algorithm of brain signals, and the commonly used classification algorithms include support vector machines, nonlinear Bayesian classifiers, artificial neural networks [28], and combinations of multiple classifiers [29]. Finally, the neurofeedback in Figure 1 can include the device state, presented stimulus, control display state, device controller state, user state and reported error, and environment state [2].

The general BCI system framework is conducive to comparing different BCI studies, and its purpose is to provide an objective method to compare BCI technology with other BCI and non-BCI user interface technologies [2].

### 2.2. Personalized BCI

Although the above general BCI system framework attempts to provide BCI benchmark applications, owing to the large individual differences between BCI users (as shown in Figure 2b), it is necessary to customize personalized BCI systems for specific BCI users on the basis of the general BCI system framework, as shown in Figure 2a.

For a personalized BCI, it is necessary to design, develop, and evaluate the BCI system based on the general BCI and fully consider the individual differences among BCI users, such as individual unique requirements, capability characteristics, mental activities, and brain signal characteristics [30]. We need to improve user satisfaction with the system, enhance the user experience, and make the system practical. Designing a personalized BCI system includes a personalized BCI paradigm [15,18] for a specific BCI user, a specific brain signal acquisition scheme suitable for the user, and a personalized brain signal processing algorithm [19,31] (including a specific preprocessing model, personalized channel selection, personalized feature extraction and selection, and a personalized classifier model). In addition, a personalized BCI system also includes the design and display of devices and control interfaces suitable for users, personalized neurofeedback, and an intelligent environment for operation [30], as shown in Figure 2c.

The above ideas of personalized BCI are also inspired by personalized medicine. Personalized medicine is a medical model that provides a patient with a personalized treatment plan according to his or her genome information and relevant individual characteristics [32]. Meanwhile, a personalized BCI is a concept and method for designing, developing, and evaluating BCI systems. Figure 2b shows that the individual differences determined by biological heterogeneity make the needs of individual BCI users different (such as communication and control with the outside world, neural regulation, and brain state monitoring). Their capability characteristics (such as self-care ability, vision, hearing, and imagination) and state characteristics (physical or physiological conditions) will affect their mental activities (including motivation), while the differences in mental activities will lead to differences in brain signal characteristics. In turn, this will lead to differences in the training decoding models, with different decoding results leading to different neurofeedback. These differences will make it difficult for the general BCI to adapt to or meet the needs of different users, which requires us to customize the personalized BCI system for specific users, as shown in Figure 2a, to achieve the transformation from the general BCI to the personalized BCI.

In the following sections, we describe the design and development methods of personalized BCIs as well as their performance evaluation and reporting criteria, especially the application of personalized BCIs.

## 3. Personalized BCI Design, Research, and Development

For a specific BCI user, to achieve the transformation from the general BCI to a personalized BCI, the design of the components of the general BCI can be personalized, as shown in Table 1. Table 1 lists the personalized research conducted for specific parts of BCIs, including the whole personalized BCI algorithm. In addition, this section also describes the brain signal acquisition scheme, specific preprocessing model, personalized neural feedback, and intelligent environment for a specific user.

### 3.1. Personalized BCI Paradigm Design

Cognitive psychology takes mental representation as the basic unit of information processing, and the construction, storage, extraction, and manipulation of mental representation are called cognitive processes, which can be reflected by brain activities [15]. A BCI infers a user’s cognitive process or state by analyzing brain activity signals. Therefore, the design and development of a BCI system can draw on the achievements of cognitive psychology.

A BCI system is mainly composed of coding and decoding, in which BCI coding is a specific mental task designed or designated in advance for the user, through which the user performs these set mental tasks to encode or “write” the user’s intention into the signal generated by the brain central nervous activity [15,38,39], such as encoding into the electroencephalogram (EEG) signal or near-infrared spectroscopy (NIRS) signal. The user’s intention is characterized by EEG or NIRS features.

BCI coding is realized by the BCI paradigm, which is based on the mental tasks set by BCI coding. The hardware and software realize the direct interaction between the brain and external devices, forming a type of communication or control system. BCI paradigm design is an important part of BCI research and development. It determines what type of stimulus users will face and chooses the target stimulus or what kind of mental activities they will actively perform, thus determining the characteristics of induced neural signals [40,41]. For example, the motor imagery (MI) BCI paradigm defines a set of MI tasks that users are required to perform in order to manipulate MI-BCI, including the limbs involved in motion and the manner of motion. The synchronous MI-BCI paradigm requires hardware and software to visually present the paradigm to users to prompt them to execute the specified MI [18] at the designed time sequence. The asynchronous MI paradigm does not require prompting.

Different BCI paradigms have different efficacies and applicability. Individual BCI users can choose paradigms according to their own needs, and, at the same time, according to their own capabilities and state characteristics, they can consider their capability to perform the mental tasks assigned by BCI paradigms and combine their own preferences for mental tasks to choose their own personalized BCI paradigms.

An important aspect of a personalized BCI paradigm is personalized mental tasks, which are evaluated by task performance (the performance of users in the tasks, such as classification accuracy), task suitability (the suitability of tasks for users), and task ease of use (the difficulty users have performing tasks). When designing a personalized BCI paradigm, it is necessary to evaluate personalized mental tasks. So far, personalized BCI paradigm design has mainly used the following two methods. One is a design method based on the mental task score [11], where tasks are recommended for users by calculating their mental task scores, and then users can choose them according to their own conditions. The other is the BCI paradigm of self-regulation based on feedback; that is, users are placed in a closed loop system and use visual presentation and other means to let them self-regulate mental tasks and their execution methods through feedback mechanisms [18].

(1) Personalized BCI paradigm design based on mental task scoring

Weyand et al. proposed a user-centered mental task selection framework. When the framework is used to select personalized mental tasks for users, multiple groups of experiments are designed. In the first several groups of experiments, users are required to perform a variety of different alternative mental tasks, and each task is scored using the proposed framework. In the next several groups of experiments, users are required to select personalized mental tasks that are suitable for them according to the scores and their feelings. At present, there are user-selected weighted slope scores (WS-US), user-selected pair-wise accuracy rankings (PWAR-US), the Repeatable Battery for the Assessment of Neuropsychological Status (RBANS) [15], and other scoring methods. For the WS-US scoring method, one can take the near-infrared spectrum acquisition method as an example to sort the tasks according to the degree of hemodynamic activity during a variety of different mental tasks, where the user selects the mental tasks according to the sorting situation. For the PWAR US scoring method, one selects appropriate classification features and uses cross-validation to perform an iteration for all possible paired combinations of candidate mental tasks, and then PWAR US will rank these task pairs from high to low. The higher the ranking, the higher the classification accuracy [15]. For the RBANS scoring method, each participant is required to select a pair of tasks in the list according to the perceived task difficulty data in all iterations [33].

(2) Personalized BCI paradigm execution based on visual feedback

Different BCI users usually perform mental tasks specified in the BCI paradigm in different ways, and the same BCI user may also perform specific mental tasks in different ways. It is necessary to find a personalized mental task execution method suitable for users to induce brain signal features with good separability, so as to further improve the performance of the BCI system, such as classification accuracy.

In addition, one of the core tenets of a BCI system is an advanced machine learning algorithm, which classifies the user’s brain signal features into desired actions or instructions. This algorithm usually operates in high-dimensional brain signal feature space. When it cannot correctly identify the desired instructions from brain signals, the reason can be found by visual feedback of data [18]. At present, the technologies that support visual feedback mainly include self-organizing maps (SOM). A SOM has the function of dimension reduction. It can visualize the recorded neural signals online into tracks on a two-dimensional SOM and provide users with real-time feedback to help them adjust the way they perform mental tasks and change their mental state in time [34,42,43].

Kuzovkin et al. used the method of visual feedback to help users choose the most suitable mental paradigm [34]. Through appropriate simplification, they could directly visualize the mental activities of users. Users could evaluate and adjust their mental activities in real time during the interaction process to achieve “what you want is what you see or what you see is what you want.” Figure 3 shows the way to provide visual feedback on the user’s mental activities to the user so that they can adjust the way they perform mental tasks in a timely manner. In the beginning, because the user has no experience imagining left-hand or right-hand movements, their way of imagining these movements is often inappropriate. The difference between the left and right brain in the visual feedback sensory motor rhythm power topographic map is not significant, and the BCI machine learning algorithm has difficulty distinguishing it. To improve the classification performance of BCI, users must adjust their own mental activity mode. With their interactive adaptation with the visual feedback system, they gradually find the most suitable left-hand or right-hand motion imagination execution mode. At this time, the visual feedback sensory motor rhythm power map has significant differences between the left and right sides of the brain, which the BCI machine learning algorithm can accurately identify.

This visual feedback method promotes the “dialogue” between the user and the BCI machine learning algorithm and visually explains to the user why the current series of mental activities is difficult for the BCI system to recognize, which mental activities can be well recognized by the system, and which mental activity execution methods should be changed. By constantly adjusting their own mental activities, users can find appropriate execution methods for each mental task to continuously improve the output performance of the BCI.

### 3.2. Brain Signal Acquisition Scheme for a Specific User

Brain signal acquisition sensor technology is one of the bottlenecks in the practical application of BCI systems. At present, the quality of the brain signals collected is not high, and the user comfort is poor [44,45]. Therefore, it is the direction of future efforts to research and develop sensors that can collect brain signals to meet the application requirements and make users more comfortable. Lyu et al. argued that the user-centered design of a brain signal acquisition scheme should ensure the safety, comfort, aesthetics, and ease of use of sensor equipment [30].

At present, there are many brain signal acquisition methods [46], but not all of them are suitable for specific BCI users. Because different BCI users have different needs, capability characteristics, and status characteristics, they will have corresponding preferences for brain signal acquisition methods. For example, some BCI users may prefer to choose wearable EEG earphones, while some BCI users may only use implantable BCIs to collect ECoG [21,22] or Spikes/LFP signals [23].

### 3.3. Personalized Processing of Brain Signals

The processing module of brain signals includes channel selection, feature extraction, and pattern recognition. Because each individual’s thinking activities and brain physiological characteristics are different, the collected brain signals will also be different. Therefore, for personalized BCI systems, personalized signal processing is a very important element.

#### 3.3.1. Specific Preprocessing Model

Brain signal preprocessing is the pre-link of subsequent feature extraction and selection and classification model construction. Its purpose is to eliminate artifacts and improve the signal-to-noise ratio. In a general BCI, a general artifact elimination model is used to improve the signal-to-noise ratio. However, the characteristics of artifacts such as electromyography and electro-oculogram caused by different BCI users are often different. It is necessary to introduce advanced machine learning algorithms to build an artifact elimination model for a specific BCI user, monitor the generated artifacts in real time, and eliminate them.

#### 3.3.2. Personalized Channel Selection

In the experimental research of BCI systems, to obtain more data, multi-channel BCI systems are often used to collect brain signals. However, not every channel contributes to classification. Some channels even reduce the classification accuracy. Different channel combinations often have different classification performances. In addition, under the same BCI paradigm or mental task, there are differences in the activation of different users’ brain regions. Therefore, it is necessary to select brain regions or corresponding channel combinations for specific BCI users to optimize the BCI system performance. At present, people can use Granger causality [47], dynamic causal modeling [48], and other means to calculate the connections between neurons and causality of information transmission. For example, in the BCI paradigm of motion imagination, the distribution of event-related desynchronization/event-related synchronization (ERD/ERS) on the scalp among users is not the same, so it is necessary to carry out personalized channel selection for different BCI users of motion imagination.

At present, there are many personalized channel selection algorithms, whose selection criteria are often different, but the main purpose is to select the most suitable channel subset for the user. For example, the sparse common space pattern (SCSP) can remove irrelevant channels, customize channel subsets for the user to produce the best classification accuracy, or reserve the minimum number of channels without affecting the classification accuracy. This algorithm has been successfully used in motion image BCIs [14]. Select distribution estimation (EDA) can capture the correlation between channels, code these relationships according to statistical correlation, and then select the subset with the best performance for BCI users [19]. In addition, Wang et al. put forward a personalized channel selection method based on a deep belief network. This algorithm is based on the idea that the channel with greater contribution to the output of the neuron has greater weight in its corresponding dimension. Thus, several channels with higher weight in the first layer of the trained deep belief network (DBN) model are selected as the optimal channel combination [31].

#### 3.3.3. Personalized Feature Extraction and Selection

To identify the mental state of BCI users with high accuracy, extracting and selecting brain signal features with good separability is the premise or basis of subsequent classification and is the core part of personalized brain signal processing. Because different users have different mental activities, abilities, and physiological conditions, the time–frequency characteristics of different users in the same channel are distributed differently under the same BCI paradigm or mental task, so personalized feature extraction and selection are required for specific users. For example, the time–frequency characteristics of ERD/ERS of different users are different for the same channel (such as C3 or C4) in the mental task of motion imagination.

At present, personalized feature extraction and selection can screen the best features for users through FBCSP, mutual information, genetic algorithms, and more. For example, Wang et al. proposed a personalized feature extraction method based on filter banks and elastic networks. First, Filter Bank Common Spatial Pattern (FBCSP) is used for feature extraction, and then the training data are used to build the elastic network logic regression model to select the best feature subset for each subject [31]. Borman et al. proposed a two-stage feature selection method that can effectively reduce the dimension of feature space. First, mutual information is used to filter out the smallest feature, and then a genetic algorithm is used in the filtered feature space to further reduce the dimension and obtain the best feature subset [13].

#### 3.3.4. Personalized Classification Model

There are many classification models of general BCIs, but the general model may not be necessarily suitable for the BCI customized for specific users. It is necessary to select the classifier type suitable for user applications for a personalized BCI system. Discrete and continuous control tasks usually select the classifier whose output is a discrete value and the regression model whose output is a continuous signal, respectively, and then use the features selected in the previous section for training to obtain personalized classification models. In addition, owing to the nonstationarity of BCI users’ brain signals (brain signal characteristics change with time), the trained classification model needs to adapt to this nonstationarity, and online adaptive machine learning can be used to update the classification model parameters.

Some studies have used transfer learning to predict the personalized classification model of subjects, using the existing data of subjects to predict the personalized model for new subjects or updating the existing model for them. Mastakouri et al. trained a personalized model for a 26th subject based on the data of the first 25 subjects in transfer learning and then predicted other experiments of the subject from this model [16]. This personalized model of transfer learning training links the EEG rhythm with sports performance, enabling researchers to deal with the heterogeneity of sports performance of different subjects. Kalaganis et al. proposed a data enhancement method for graph signals. This method uses graph variant empirical mode decomposition to generate artificial EEG signals to improve the classification accuracy of personalized BCI [36].

Owing to the large difference among brain signals between individuals and the change of brain signal characteristics over time, the effect of transfer learning for BCI is limited. Although researchers have proposed some personalized classifier models, the work of Lopes Dias et al. [49] shows that, for some BCI users, the classification effect of the personalized classifier model is not significantly different from the general classifier model, or even lower than that of the general classifier model, which may be caused by the training of the personalized classification model.

#### 3.3.5. Personalization Algorithm Design Based on the Overall BCI System

In addition to the personalized design of the BCI paradigm, brain signal acquisition scheme, preprocessing model, channel selection, feature extraction and selection, and classification model, it is also necessary to customize the BCI system for specific BCI users from the perspective of the whole BCI system and further optimize it.

Ugarte et al. proposed a weighted discriminator (WD) index [35] which reflects the preference of subjects for corresponding parameters (such as different signal acquisition and processing schemes), selects the most suitable scheme for subjects, and integrates it into their customized BCI system. Bashashati et al. proposed an optimization algorithm based on Bayes and applied it to the BCI of motion imagination [37]. According to the brain signal characteristics of each subject, the algorithm uses Bayes to optimize parameters such as channel, frequency band, and time period, and evaluate these parameters to provide a personalized BCI system. The above research is based on the entire BCI system to design personalized algorithms, and the research in this area needs to be further developed.

### 3.4. Personalized Neurofeedback

Neurofeedback (NF) is a biofeedback technology based on central nervous activity [50,51]. Neurofeedback is one of the key links of the BCI system. It forms a closed loop bi-directional BCI system, feeds back neural activity to users in real time through visual, auditory, or tactile feedback forms, and uses the principle of operant conditioned reflex to enable users to learn to independently enhance or inhibit neural activity. Endogenous regulation of brain activity is achieved through neural feedback training, as shown in Figure 1. Neurofeedback is one of the key links of the BCI system, which forms a closed-loop bidirectional BCI system, as shown in Figure 1. Through neurofeedback, the user’s brain activity characteristics, BCI decoding results, and results of communication or control with devices can be visually fed back to the user in visual, auditory, or tactile ways to adjust their mental activity mode, so as to adjust their brain signals, drive the BCI system, and improve its performance [2]. BCI operation usually requires effective interaction between two adaptive controllers (user and BCI adaptive algorithm), and neurofeedback plays a key role [1].

Because different BCI users have different capabilities, states, and ways to perform the same mental tasks, their brain signal characteristics are also different, and thus, the BCI decoding results and communication or control results are different, which ultimately leads to different contents of the feedback signals to users, and their self-regulation processes are also different. In addition, different BCI users have preferences for the form or interface of neurofeedback. Therefore, it is necessary to design personalized neural feedback for specific BCI users.

Personalized neurofeedback should not only adjust the neural activities of specific users but also improve the user’s sense of experience and satisfaction in the process of feedback [52,53]. However, the existing BCI neurofeedback is relatively brain-intensive, and users are easily tired mentally and physically, and even bored. To increase the effectiveness of neurofeedback, a specific feedback channel should be designed according to the characteristics of the user (such as health, hearing, or visual impairment) and let them conduct neurofeedback easily, rather than completing a tedious task.

### 3.5. Personalized Intelligent Environment

The BCI system operates in the environment, and its users interact with the environment. The performance of the system is closely related to the operating environment. At present, most BCI systems have been developed in well-controlled and structured laboratory environments. Such BCI systems are usually difficult to adapt to the daily life and working environment of users [30], Some studies have introduced shared control to improve the robustness of BCI systems [54]. In addition, the performance and efficacy of the current BCI systems are still very limited, so it is necessary to design an intelligent operating environment. For example, an intelligent environmental system can be built by technologies such as intelligent sensor networks and deep learning in the operating environment and integrated with the BCI system to further improve its performance of the BCI system.

According to the needs of specific BCI users (the efficacy provided by BCI is required to solve their problems) and their capabilities, and taking full account of their living or working environment (which may have some uniqueness), an intelligent environment suitable for the operation and operation of a specific BCI system (such as different BCI paradigms) can be built, considering user preferences for the intelligent environment. Designing such a personalized intelligent environment is conducive to enhancing the interaction performance between the BCI system and users and improving the user’s sense of experience and satisfaction [55,56].

## 4. Evaluation of Personalized BCI Performance

To compare the performance of different BCI systems, the criteria for reporting their performance are crucial [2], and personalized BCI is no exception (as shown in Figure 2c). In addition, the evaluation of personalized BCI performance methods also helps specific users to evaluate such BCIs to promote their improvement.

The personalized BCI is designed and developed on the basis of the general BCI (as shown in Figure 2a). Therefore, the performance of the personalized BCI also needs to be evaluated quantitatively or objectively according to the general BCI performance indicators. In addition, it needs to be evaluated from the perspective of specific BCI users through indicators such as satisfaction [30,54].

### 4.1. Performance Evaluation Methods for General BCI System

A general BCI system performance evaluation method is shown in Table 2, including multiple quantitative indicators from the perspective of researchers. As shown in Table 2, the most commonly used indicators are classification accuracy, the information transmission rate, and Cohen’s kappa. These three indicators reflect the accuracy, transmission rate, and consistency of BCI, which can be evaluated to a certain extent, while sensitivity, specificity, noise factor, F-measurement [2], and other indicators can be used as supplements to these indicators.

### 4.2. Evaluation Method of Personalized BCI Performance

As shown in Figure 4, the satisfaction evaluation of specific users regarding BCI sensors is extremely important, because the performance of BCI sensors will seriously affect user acceptance of BCIs [30,54], The workload evaluation of a specific user controlling the BCI will also affect the user’s acceptance of BCI [30]. In addition, Figure 4 also includes a user-specific visual analog scale [2,58] and an evaluation of a specific user’s overall satisfaction with BCI [30,54]. The latter can use Quest 2.0 and its extension form of user satisfaction with auxiliary technology [56,58,59]. In particular, interview/follow-up evaluation with specific users is necessary and important, in which four issues need to be considered when BCI technology is converted into practical application [1].

## 5. Personalized BCI Application

Whether it is the personalization of an experimental paradigm, brain signal processing, or neural feedback, the ultimate goal is to build a BCI system platform and put it into practice. In this section, we will discuss the applications of personalized BCIs in the rehabilitation of motor dysfunction, psychiatric treatment and rehabilitation, emotion recognition, and other fields. Table 3 lists the main personalized BCI applications so far, and this section will review the contents in the table.

### 5.1. Application of Personalized BCI in Auxiliary Control

At present, BCIs are most widely used in auxiliary control, such as communication and control, and in motion replacement.

(1)Communication and control

The basic principles of BCI communication and control are similar, both of which realize the output of external instructions/symbols by identifying specific patterns of brain signals. Through BCIs, users with severe disabilities can communicate with others and control the external device. Abiri et al. designed a social robot based on gesture control [60]. By combining this with the neural feedback mechanism, they customized the decoding model for users to control the gestures of the social robot. Uma et al. developed a personalized GUI application that collaborated with the EEG device to access the user’s neesd [43]. Abiri et al. confirmed that in cursor control, there is a positive correlation between individual visualization ability and the controllability level of the cursor, which can provide research directions for personalized cursor control [70].

(2)Motion replacement

Annalisa Colucci et al. mentioned in the literature published in 2022 [71] that the brain/neural exoskeleton (B/NE) will play a key role in improving the effectiveness of personalized treatment strategies. Coscia M et al. [61] adjusted and improved the control parameters of B/NE training according to the patient’s individual ability by monitoring the physiological biomarkers that predict mental exhaustion, such as heart rate variability, galvanic skin response, or respiratory rate. In addition, Vinoj et al. developed a brand-controlled lower limb exoskeleton that can be customized according to the degree of disability [62].

### 5.2. Application of Individualized BCI in the Rehabilitation of Neurological Diseases

The following takes the treatment and rehabilitation of Parkinson’s disease (PD) and stroke as examples to discuss the application of personalized BCIs in the rehabilitation of motor and cognitive dysfunction.

(1)Treatment and rehabilitation of Parkinson’s disease

At present, there is no way to completely cure PD [72], and personalized BCIs are expected to improve the symptoms of Parkinson’s disease patients. Bronte-Stewart et al. proposed a personalized dual threshold control strategy using the bidirectional deep brain–computer interface (dBCI) and applied the strategy to neural or motor input to drive closed-loop subthalamic nucleus deep brain stimulation (STN-DBS) for the treatment of PD [63]. This is a personalized BCI. Different from traditional BCI, it seamlessly adjusts the parameters of nerve stimulation according to the activity status and medication cycle of specific users. The purpose is to provide specific Parkinson’s patients with the parameters of the best treatment and rehabilitation effects. This study demonstrated the feasibility and effectiveness of closed-loop DBS for PD for the first time.

(2)Rehabilitation after stroke

Strokes lead to cognitive disorders or/and motor disorders in patients [73]. For the rehabilitation of these disorders, personalized BCIs mainly use BCI-based neurofeedback training and transcranial electrical stimulation (TES) to intervene.

(a)Rehabilitation of cognitive impairment

A study found that increasing the energy of a specific EEG frequency band can improve cognitive performance. Kübler et al. used this discovery to design a neural feedback training module based on BCI to enhance the cognitive function of stroke patients and proposed a neuropsychology algorithm. According to the neuropsychology test results, different suitable neurofeedback training modules can be customized or allocated for specific stroke patients [64].

(b)Rehabilitation of dyskinesia

Mane et al. used transcranial direct current stimulation coupled with BCI (TDCS-BCI) to intervene in the upper limb motor disorder of stroke patients. They found that the brain symmetry index and power ratio index (PRI) were the best predictors of TDCS-BCI intervention, and these predictors were helpful for identifying the biomarkers of different patients [65]. The biometric markers of specific patients can be used to predict their expected response to existing interventions, and the interventions with the highest expected benefits can be recommended to patients to achieve personalized rehabilitation programs. Compared with the traditional BCI, this personalized BCI application solves the adaptability of the general scheme to different patients and provides each patient with a rehabilitation scheme that is suitable for them and has higher expected benefits.

### 5.3. Application of Individualized BCI in the Rehabilitation of Mental Disorders

Output-type BCI, which mainly outputs instructions from the brain, mainly realizes communication and control with the outside world. Compared with this type of BCI, there is another type of BCI that mainly uses external devices or machines to bypass the nerve or muscle system to directly input electricity to the brain (such as deep brain stimulation (DBS)), transcranial direct/alternating current stimulation (tDCS/tACS) [74,75,76], magnetism (such as transcranial magnetic stimulation (TMS)), sound (such as transcranial ultrasonic stimulation (TUS)), and light stimulation or neurofeedback (input BCI) to regulate central nervous activity. Such a BCI can be used for physical intervention in mental disorders to promote rehabilitation.

One of the main factors of the regulatory effect of input BCIs on mental disorders is the optimization of the optimal neuromodulation or stimulation parameters for a specific patient, that is, personalized stimulation parameter settings. Fellous et al. used advanced machine learning algorithms to identify brain states and optimized stimulus parameters using neural features. They introduced explainable artificial intelligence (XAI) to identify specific biometric markers (such as event related potentials (ERPs) [76,77,78,79]) to detect abnormal neural activity [80]. For example, delayed and/or reduced ERP amplitudes can be observed in alcohol-addicted patients and animal models [78,79,81,82]. When abnormal neural activity is detected, the stimulation to the brain is turned on, the stimulation parameters are adjusted adaptively, and the stimulation is turned off immediately after the normalization of brain activity [66]. ERPs and machine learning can support the diagnosis of mental symptoms and predict the disease progression and treatment results of specific subjects so as to achieve personalized treatment and rehabilitation [67,83].

### 5.4. Application of Individualized BCI in the Rehabilitation of Mental Disorders

Emotion regulation is very important for an individual’s physical and mental health, and emotion recognition is the basis of emotion regulation. The emotions of different individuals will change with time and environment [84]. Therefore, BCI needs to be customized for specific individuals to identify emotions.

Affective BCI (aBCI) monitors emotional states by measuring neurophysiological signals, helps users actively customize mental tasks, and improves the performance of human–computer interactions [68]. For example, in one study individuals used emotional changes to control a game. When they realize that their emotional state will affect the game parameters, they will actively change their emotions according to their preferences, adjust their mental tasks through the results of the game manipulation and emotional feedback, and achieve personalized aBCI-based game manipulation [85].

In addition, Daly et al. developed an emotional state detection system for brain–computer music interfaces (BCMI) [69]. Among them, BCMI induces users’ emotions through music to help them regulate their emotions [37]. Daly et al. trained the classifier for each subject (the music stimulus used in the experiment was created by the affective algorithm composition (AAC) system [86,87]). The experimental results showed that, compared with the ordinary BCIM (*p* < 0.05), the personalized BCMI system could more accurately detect the subjects’ emotions (*p* < 0.01).

### 5.5. BCI-Related Research Considering Specific Users

MI, P300, and SSVEP are three typical BCI paradigms. Owing to the large differences in the needs, capabilities, and brain signal characteristics of specific BCI users, it is a current and future research direction to customize the above BCIs for specific users on the basis of general BCIs.

In an MI-BCI study, Delisle Rodriguez and others automatically located the band with the highest power during the movement imagination of specific subjects through the sparsity constraint and total power used for time–frequency representation to improve classification accuracy [88]. Furthermore, Wu et al. proposed a discriminative and multi-scale filter bank tangent space mapping (DMFBTSM) algorithm. The algorithm can customize filter banks for specific subjects to identify multiple MI tasks [89]. In addition, Kumar et al. optimized time domain filters for specific subjects [90], and Gaur et al. used the Pearson correlation coefficient to select channels for specific subjects [91].

In a P300 BCI study, Sellers et al. optimized the P300 BCI system for specific users by customizing parameters such as matrix size and inter stimulus interval (ISI) [92]. Erdogan et al. analyzed the response of specific users to the spelling paradigm and determined the most appropriate P300 detection band for them [93]. In recent years, Wang et al. used a multi-scale convolutional neural network (MS-CNN) to train a general decoding model and then adjusted the general model by using part of the data of specific subjects through transfer learning to obtain a customized decoding model [94]. In addition, Li et al. proposed a TrAdaBoost algorithm based on cross-validation and an adaptive threshold (CV-T-TAB). By selecting and combining existing classifiers of multiple subjects who perform well on new subjects, the amount of data required for training is reduced, and the classifier performance of specific subjects with a small amount of data is effectively improved [95].

In SSVEP research, Ravi et al. customized channel subsets for specific users to reduce the impact of changes in stimulus spacing on SSVEP decoding performance [96]. Meanwhile, Rejer et al. used wavelet transforms (WT) to determine the optimal flicker frequency for a specific user to achieve customization of SSVEP-BCI [97]. In addition, Mehdizavareh et al. used the training data of other subjects to optimize the super-parameters of the canonical correlation analysis (CCA) model for specific subjects [98]. Peters introduced a method to adaptively select the test length of a specific user, which can improve the information transmission rate and the accuracy of letter selection [99]. The above research has achieved personalized design for specific subjects or users in some aspects of SSVEP-BCI.

## 6. Challenges and Prospects of Personalized BCI

### 6.1. Challenges Faced by Personalized BCI

The challenges faced by personalized BCIs not only come from the limitations of general BCIs, but also have further problems to be solved. The following section will describe the problems that need to be solved when customizing BCI for specific users from the perspective of the transformation from general BCI to personalized BCI, as well as BCI paradigm, sensors, brain signal analysis, neurofeedback and evaluation methods.

(1)How to personalize the general BCI to suit specific users

In the research on personalized BCI, how to deal with the relationship between personalized BCI and the general BCI, that is, how to make personalized designs for specific users based on the general BCI to satisfy them, is a challenge that needs to be overcome to move BCI into practical applications, and is an important direction of future BCI research.

(2)Which BCI paradigm suits or satisfies a specific user

Although the current personalized BCI research allows specific users to freely choose existing paradigms [11] and automatically adjust the established paradigm through neural feedback [18], the existing paradigms (such as the MI paradigm, P300 paradigm, and SSVEP paradigm) have inherent defects. For example, Chuan-Chih Hsu et al. proposed that low-frequency SSVEP would lead to the risk of photosensitive epilepsy [100]. The P300 experiment took too long, and MI had high requirements of user imagination. The existing paradigms have difficulty meeting the needs of specific users. Another challenge and important direction of personalized BCI research is how to improve the existing BCI paradigm for specific users or how to customize a new BCI paradigm for them and incorporate user evaluation indicators into the paradigm design.

(3)Which brain signal acquisition sensor is suitable for satisfying a specific BCI user

The limitation of brain signal acquisition sensor technology is the bottleneck hindering the practical application of BCI systems. The current non-invasive BCI sensor user comfort (such as wet electrodes needing gel or normal saline to reduce the impedance and dry electrodes needing a certain pressure to keep it close to the scalp) and ease of use are poor (placing or wearing the electrode usually requires assistance from others). The safety of invasive BCI sensors also has problems (such as trauma caused by ECoG recordings and intracortical recordings). These factors mean the user satisfaction of BCI systems is low. Yu and Qi conducted a consumer survey in 2018 [101] to select the best wearable non-invasive EEG BCI. The three main characteristics of selecting appropriate earphones are as follows: safety 84.26%, effect accuracy 59.34%, and comfort 58.3%. This proves that one of the challenges facing personalized BCI is designing sensors that can satisfy specific users on the premise of safety and collecting effective brain signals.

(4)Which brain signal characteristics of a specific user are suitable for driving a BCI

When the general BCI model trained by the brain signal data of many subjects or users is used for a specific user, it is usually difficult to ensure that its performance meets the requirements, because the brain signals vary greatly between individuals and within an individual over time. Under the specific BCI paradigm, how to meet the needs of specific users and customize feature subsets and classification models to better drive BCIs requires further research. For example, Qi in 2021 proposed a multilayer Recurrent Neural Network (RNN) consisting of a Long Short-Term Memory (LSTM) module and a dropout layer [102]. It effectively improved the classification performance of brain signals and showed strong anti-interference ability.

(5)What kind of neural feedback can improve BCI performance and satisfy specific users

Neurofeedback in BCI refers to the brain activity information of a specific user or the communication and control results obtained from it, so as to help users adjust their mental activity strategies and improve the performance of controlling the BCI. It is also important for future personalized BCI research to investigate how to customize the neural feedback scheme for a specific user (including the characteristics of brain signals to be fed back, the direction of regulation, and the presentation mode of feedback) to avoid boring content and forms of neural feedback and provide neural feedback with motivation, immersion, and user satisfaction. For example, D. Borton et al. proposed that the specific needs of patients can be met through closed-loop neural feedback, adjustment of parameters and the introduction of different neural regulation features, and the opportunity to alter the dynamic state of neural networks can be provided [103].

(6)How to evaluate the performance of BCI customized for specific users

Although there are some criteria for evaluating or reporting BCI performance [57], mainly from the technical perspective, BCI is directly controlled by a specific user’s brain signals to improve the user’s work efficiency and quality of life. Therefore, it is also necessary to combine user-centered BCI evaluation methods to develop a BCI that the user is satisfied with. Although this paper tries to provide an evaluation method of personalized BCI, it still needs to be further improved and verified.

### 6.2. Limitations of Personalized BCI

Although a personalized BCI has many advantages and ways to be improved, it has its own limitations due to it needing to be customized for specific users.

First of all, there is the problem of funds. Specific users may not be able to accept the soaring production costs of customized personalized BCIs. At the same time, the research and development of personalized BCI also needs funding. It is difficult to obtain financial support for personalized research and development only for specific users. Secondly, the research time of BCI systems is long, which makes it difficult for specific users to use personalized BCIs quickly. In the long customization process, the needs of users will gradually change, which makes it unable to really meet the needs of specific users. Finally, some users do not have the conditions to use BCIs or have poor abilities to use BCI [104], so it is difficult to customize personalized BCIs for these users.

### 6.3. Future of Personalized BCI

After years of development, BCIs have become general systems that have potential applications in medical and non-medical fields and which have been verified by many studies. At present, there is still a big gap between BCI research and practical applications. The research on personalized BCIs is mainly concentrated in the field of medical rehabilitation, focusing on enabling patients to obtain the most suitable treatment parameters. In the future, it is expected to achieve the one person, one program paradigm, and combine BCIs with other methods, not only to adjust the paradigm and treatment parameters, but also to enable patients to obtain a complete set of their own treatment programs from treatment to rehabilitation. Personalized BCIs can also have their own applications in other fields, such as life, games, military, and transportation. In the field of life, personalized BCIs may develop into multi-mode wearable devices and provide users with the most suitable services according to long-term and continuous brain signals (such as monitoring their physiological status at any time and customizing treatment plans for them to a certain extent). This application may greatly facilitate the life of the disabled or the elderly. In the field of transportation, personalized BCIs can be combined with automatic driving technology to customize the route and driving style for users through their characteristics. In the military field, personalized BCIs can customize combat plans for soldiers through their individual characteristics (for example, adjust their shooting distance and intensity for soldiers through special guns).

At present, the research on personalized BCIs is developing in two directions. On the one hand, we can trace the signal to specific neurons and explore the physiological differences of different users through the personality connection between nerves. On the other hand, we can customize personalized sensors for users to achieve personalized recognition of user intentions. Personalized BCIs will be one of the important ways for general BCIs to become practical. They will customize BCIs for specific users on the basis of general BCIs to meet user needs and improve satisfaction. The concept and method of personalized BCIs can promote the transformation of the BCI industry.

## 7. Conclusions

This study first attempted to give a definition of personalized BCIs, and describes the design and development of personalized BCIs, including the personalized BCI paradigm, channel selection, feature extraction and selection, classification model and neurofeedback. Then, we combined the general BCI system performance evaluation method with the evaluation method from the user’s perspective, and discussed the evaluation method of personalized BCI. Thirdly, we introduced the application research of personalized BCIs, including neuropsychiatric rehabilitation and emotion recognition, and finally discussed the challenges and prospects of personalized BCI. Due to the individual differences between different users of BCI, the design and development of personalized BCIs will be very important development directions in the future. How to meet the needs of specific users, improve the key technologies in personalized system design, and establish a sound evaluation system will be the key directions in future research.

## Figures and Tables

**Figure 1 jpm-13-00046-f001:**
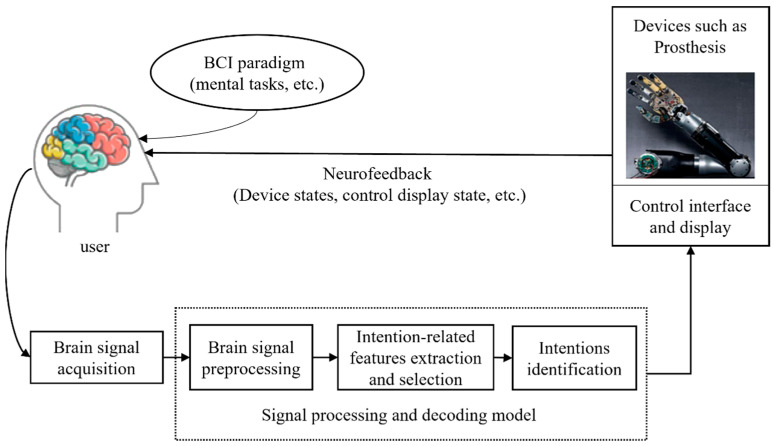
Schematic diagram of a general BCI system.

**Figure 2 jpm-13-00046-f002:**
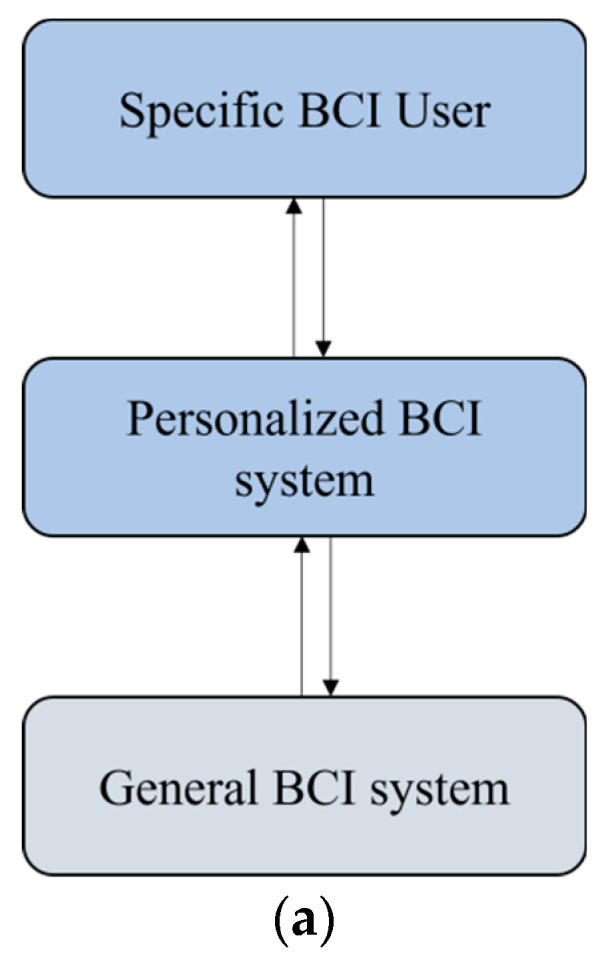
(**a**) Individual differences between BCI users; (**b**) developing and designing personalized BCIs based on general BCIs; (**c**) personalized BCI system diagram.

**Figure 3 jpm-13-00046-f003:**
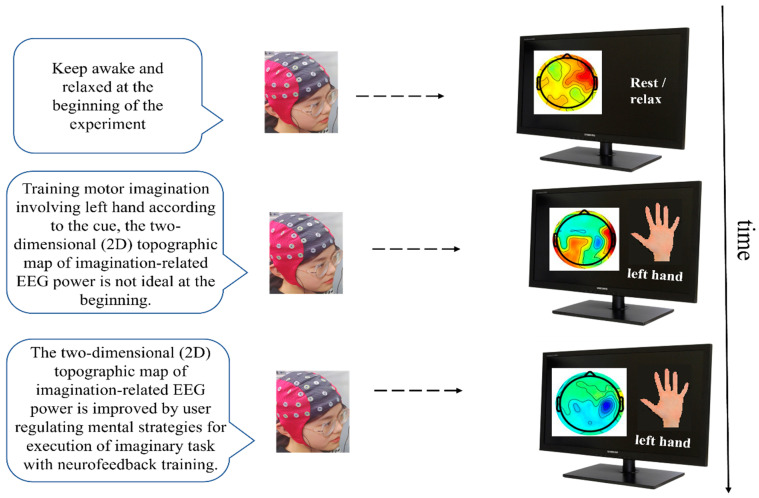
Way of providing visual feedback to users to make them adjust their mental tasks in a timely manner.

**Figure 4 jpm-13-00046-f004:**
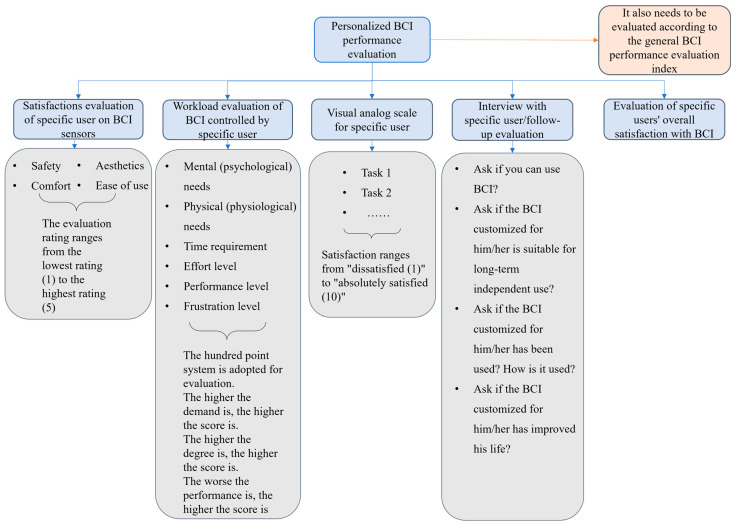
Evaluation method of personalized BCI performance. The general BCI performance evaluation index comes from Table 2.

**Table 1 jpm-13-00046-t001:** Studies on personalized BCI design.

Reference	Specific Works for Personalized BCI	Personalized Component
Weyand et al., 2015 [15]	Proposes the use of personalized mental tasks; explores a user-centered approach Proposes user-selected weighted slope scores (WS-US) and user-selected pair-wise accuracy rankings (PWAR-US)	Personalized BCI paradigm
Yeo et al., 2018 [33]	Proposes the Repeatable Battery for the Assessment of Neuropsychological Status (RBANS)	
Kuzovkin et al., 2019 [34]	Realizes visualization of mental state space for personalized BCI	
Wang et al., 2021 [31]	A classifier-based approach for personalized channel selection in deep belief networks is proposed	Personalized channel selection
Astigarraga et al., 2016 [19]	Presents a brain–computer interface for motion imaging based on a distributed algorithm for EEG channel selection	
Arvaneh et al., 2011 [14]	Proposes a sparse common spatial pattern (SCSP) algorithm for brain signal channel selection	
Ugarte et al., 2017 [35]	An algorithm model is proposed to select different types of brain signal processing methods for subjects by predicting their wishes	Personalized feature extraction and selection
Bormane et al., 2009 [13]	A feature selection method is proposed to effectively reduce the dimension of multi-channel feature space	
Wang et al.,2021 [31]	A personalized feature extraction method based on filter banks and elastic networks is proposed	
Kalaganis et al., 2020 [36]	A data enhancement method for graph signals is proposed to improve the accuracy of personalized classification	Personalized classification model
Mastakouri et al., 2017 [16]	Proposes a personalized model based on a transfer learning algorithm for the rehabilitation of patients with motion disorders	
Bashashati et al., 2016 [37]	Uses a Bayesian-based optimization algorithm to adjust parameters	The whole personalized BCI algorithm
Ugarte et al.,2017 [35]	An algorithm model is proposed to select different types of brain signal processing methods for subjects	

**Table 2 jpm-13-00046-t002:** Performance evaluation methods for general BCI systems.

Index	Computing Method	Explanation
Classification positive accuracy	$p=\frac{\sum C_{i,i}}{N}$	p is the correct classification rate; Ci,i is the ith element of the diagonal of the confusion matrix; N is the total number of trials
Information transmission rate	$\mathrm{ITR}=\frac{1}{T}\left[60\left(\log_{2}N+P\log_{2}P\;+\left(1-P\right)\log_{2}\frac{1-P}{N-1}\right)\right]$	ITR is the amount of information transmitted in unit time, N is the target number, P is the accuracy rate, and T is the time required to output an instruction
Cohen’s kappa	$\kappa =\frac{p-p_{0}}{1-p_{0}},{\;p}_{0}=\frac{\sum C_{i,:}C_{:,i}}{N^{2}}$	Κ is the consistency indicator between nominal scales, Ci,: and C:,i are row i and column i of the confusion matrix, respectively, and N is the total number of trials
Sensitivity	$\mathrm{Se}=\frac{\mathrm{TP}}{\mathrm{TP}+\mathrm{TN}}$	Se is sensitivity, TP is true positive, TN is true negative
Specificity	$\mathrm{Sp}=\frac{\mathrm{TN}}{\mathrm{TN}+\mathrm{TP}}$	Sp is specific, TP is true positive, TN is true negative
Noise factor	$F=\frac{\mathrm{FP}}{\mathrm{TP}+\mathrm{FP}}$	F is false positive, FP is false positive, TP is true positive
F-measurement	$F_{\alpha }=\frac{\left(1+\alpha \right)\cdot \left(1-F\right)\cdot \mathrm{Se}}{\alpha \cdot \left(1-F\right)+\mathrm{Se}}$	$F_{\alpha }$ is a measure of classifier performance under different significance levels α, F is the false detection rate, Se is sensitivity
Failure rate [30]	$\lambda =\frac{M}{\Delta t\cdot N}$	λ is the failure rate, M is the number of failed BCI products during the working time, N is the total number of BCI products, and Δt is BCI working time
Mean time between failures [30]	$\mathrm{MTBF}=\frac{1}{\lambda }$	MTBF is the mean time between failures; λ is the average failure rate
Fitts Throughput [57]	$\mathrm{TP}=\frac{\mathrm{ID}}{\mathrm{MT}}$	TP is throughput; ID is index of difficulty; MT is mean movement time
Receiver Operating Characteristic Curve(ROC) [57]	$\mathrm{TPR}=\frac{\mathrm{TP}}{P}\;\mathrm{FPR}=\frac{\mathrm{FP}}{N}$	The abscissa of the curve is the false positive rate (FPR). N is the number of real negative samples, and FP is the number of positive samples predicted by the classifier among N negative samples. The ordinate is the true positive rate (TPR). P is the number of real positive samples, and TP is the number of positive samples predicted by the classifier among the P positive samples.
Area Under Curve (AUC) [57]	Calculated by mean and variance	Area under ROC curve, generally between 0.5 and 1

**Table 3 jpm-13-00046-t003:** Different applications of personalized BCIs.

Reference	Applications	Direction
Abiri et al. 2017 [60]	Designed a social robot based on gesture control.	Auxiliary Control
Uma et al. 2017 [43]	Developed a personalized GUI application that collaborated with the EEG device, accessed the user’s needs,	
Coscia M et al. 2019 [61]	adjusted and improved the duration and intensity of brain/neural exoskeleton (B/NE)training according to the patient’s individual ability.	
Vinoj et al., 2019 [62]	Developed a brain-controlled lower limb exoskeleton, and customized it according to the degree of disability to assist users with lower limb disorders in rehabilitation training.	
Bronte-Stewart et al., 2020 [63]	A personalized dual threshold control strategy was proposed to drive closed-loop subthalamic nucleus deep brain stimulation (STN-DBS).	Parkinson’s
Kübler et al., 2017 [64]	A neuromental algorithm was developed to assign different neural feedback training modules to different stroke patients.	Cerebral apoplexy
Mane et al., 2019 [65]	Used biometric markers to predict patients’ expected responses to existing interventions and provide patients with high expectations.	
Parastarfeizabadi et al., 2017 [66]	Adaptive adjustment of stimulus parameters.	Rehabilitation of mental disorders
Campanella.,2013 [67]	ERPs and machine learning can help predict the disease progression and treatment results of specific subjects.	
He et al., 2020 [68]	Personalized tasks through emotional reflex control and automatically modified the human–computer interaction process.	Emotional recognition
Daly et al., 2017 [69]	Developed a high-performance emotional state detection system.	

## Data Availability

Not applicable.

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
