# Peer review of "Personalized Brain–Computer Interface and Its Applications"

_jpm, 2022, doi:10.3390/jpm13010046_

Round 1
Reviewer 1 Report
This paper summarizes the research progress on the critical scientific and technical issues involved in personalized BCI. The authors analyzed personalized BCIs, their design, development, evaluation methods, and applications. This topic is meaningful to the personalized brain–computer interface (pBCI), and specific BCI user. However, there are some issues that should be addressed.
i. Some sentences in the abstract are repetitive. It is suggested to reorganize the abstract writing.
ii. It is recommended to present in the first section so that it can highlight the specific scope of this article. The meaning of the assessment experiment should be highlighted.
iii. The literature review is not thorough about the application and the contributions. To highlight the contributions, it suggests reorganizing the section of the related work with real applications. It is recommended to read more related works and consider discussing their application scenarios in the introduction and discussion, for example, A Multimodal Wearable System for Continuous and Real-time Breathing Pattern Monitoring During Daily Activity.
iv. The contribution of this paper is not enough. The theoretical contribution is not enough and the experimental validation is clear. It suggests revising the contributions section and making these points clear and strong.
v. The quality of the figure is not enough for a high-quality paper. It suggests being easier and clear for the reader to recognize the objectives of each figure.
vi. There should be a further discussion about the limitation of the current works, in particular, what could be the challenge for its related applications. To let readers better understand future work, please give specific research directions, for example, Multi-sensor guided hand gesture recognition for a teleoperated robot using a recurrent neural network.
Overall, proofreading is preferred. The current version is not written in a good and clear way. The English description should be improved, and the grammar should be carefully addressed. It suggests consulting the publisher to polish your English writing, for example, http://mugepaper.com or using https://www.ajjjesss.com/ or other similar services. There are no other comments before publication.
Reviewer 2 Report
“Personalized Brain–Computer Interface and Its Applications”
Overall strengths of the article:
This manuscript discussed the numerous problems associated with personalized Brain-computer interfaces (BCIs) to provide practical applications of personalized BCI. In this review, the authors defined and explained personalized BCI. Personalized BCI design, development, evaluation methods, and applications are discussed in detail. Finally, the challenges and future directions of personalized BCI are summarized. This study is expected to provide some valuable ideas for innovative studies and practical applications of personalized BCI. The brain signals of different users are variable, and there are differences in user needs and capability characteristics. It is necessary to customize personalized BCI for specific BCI users. An interesting study, the structure is good, and the figure is clear. I have some major suggestions. Details are in the specific comments section.
Specific comments on weaknesses:
Major concerns:
1. In general, there are general statements but no references. Throughout the whole manuscript, many statements were made without proper reference citing, I strongly request to provide proper citations.
2. Reference citations are not consistent throughout the whole manuscript please be consistent when citing a reference. Like; “BCI has potential medical 1 and non-medical applications. So far, researchers have conducted many studies on BCI [1,2], proposed a general BCI system 3, and expected it to have a wide range of applications. However, BCI has poor user experience and satisfaction. BCI is ultimately driven or controlled by brain signals induced by a specific user’s mental activities. The characteristics of different users’ mental activities and brain signals induced by them vary greatly 4”.
3. Line 73- 81; “For a personalized BCI, it is necessary to design, develop, and evaluate the BCI system based on the general BCI and fully consider the individual differences among BCI users, such as individual unique requirements, capability characteristics, mental activities, and brain signal characteristics, so as to improve user satisfaction with the system, enhance the user experience, and make the system practical”. “Designing a personalized BCI system includes a personalized BCI paradigm for a specific BCI user, a specific brain signal acquisition scheme suitable for the user, and a personalized brain signal processing algorithm (including a specific preprocessing model, personalized channel selection, personalized feature extraction and selection, and a personalized classifier model)”. I strongly suggest breaking long sentences like these into small ones for better clarity and understanding. Please provide references too.
4. A regress English edits are needed throughout the manuscript.
5. Limitations of the personalized BCI should be listed before the conclusion.
Minor points:
1. Line 241: EOG: is Electrooculography not ophthalmic electricity (EOG)
2. Better to provide a separate list of abbreviations used. I found some of these abbreviations used in the text were not defined. Like, event-related potentials (ERP): event-related desynchronization, ERD & event-related synchronization, ERS needs to be explained. Similarly, FBCSP?
3. References in the tables need formatting.
Author Response
请参阅附件。

Reviewer 3 Report
1. Each step of a general BCI system should be introduced in a more detail way, since there are a lot of related research for each step of a general BCI system. For example, for the signal processing and decoding model, you can refer to Toward Open-World Electroencephalogram Decoding Via Deep Learning: A Comprehensive Survey.
2. 63 references are not enough for a survey paper. More related references should be added in this paper.
3. This paper should cover more BCI related applications, and sum the related works and methods up in this paper
4. More performance evaluation methods should be added, such as the AUC and FPR in the EEG-Based Seizure Prediction via Model Uncertainty Learning.
5. Changes in this area should be introduced in a more scientific and deeper way.
6. The conclusion of this paper should be rewritten.
7. There are some grammatical problems in the paper, please check and edit.
Round 2
Reviewer 2 Report
In the revised manuscript authors have successfully addressed all the comments raised by the reviewer and incorporated suggestions to improve the quality of the paper. I think this manuscript has been improved from the previous version.
Reviewer 3 Report
I suggest accepting this paper.